# SuperPos-Prompt: Enhancing Soft Prompt Tuning of Language Models with Superposition of Multi Token Embeddings

## Abstract

Soft prompt tuning techniques have recently gained traction as an effective strategy for the parameter-efficient tuning of pretrained language models, particularly minimizing the required adjustment of model parameters. Despite their growing use, achieving optimal tuning with soft prompts, especially with smaller datasets, remains a substantial challenge. This study makes two contributions in this domain: (i) we introduce SuperPos-Prompt, a new reparameterization technique employing the superposition of multiple pretrained vocabulary embeddings to improve the learning of soft prompts. Our experiments across several GLUE and SuperGLUE benchmarks consistently highlight SuperPos-Prompt's superiority over *Residual Prompt* tuning, exhibiting an average score increase of $+6.4$ in *T5-Small* and $+5.0$ in *T5-Base* along with a faster convergence. Remarkably, SuperPos-Prompt occasionally outperforms even full fine-tuning methods. (ii) Additionally, we demonstrate enhanced performance and rapid convergence by omitting dropout from the frozen network, yielding consistent improvements across various scenarios and tuning methods. Unlike many existing strategies, our approach does not rely on the availability of a proficient pretrained source prompt for initialization, thereby ensuring notable flexibility and more effective combination of related prompt candidates.

## 1 Introduction

Optimizing deep neural network models generally requires substantial data to achieve optimal performance. This prerequisite has underscored the importance of transfer learning in various domains of deep learning, including natural language processing (NLP) (Ruder et al., 2019), computer vision (Gopalakrishnan et al., 2017), and reinforcement learning (Zhu et al., 2023). Transfer learning is an approach in which a pre-trained model is adapted and fine-tuned for new tasks, particularly when labeled data is limited. Foundation models, denoted as Large Language Models (LLMs) in NLP, are large models trained on vast datasets utilizing self-supervised methodologies (Pfeiffer et al., 2023) acting as a base for further fine-tuning on new tasks. Over time, the scale of publicly available LLMs has remarkably grown, from BERT's 340 million parameters (Devlin et al., 2019) to contemporary models housing up to 100 billion parameters (Almazrouei et al., 2023).

Full fine-tuning of models is one approach to overcoming the challenges posed by limited data at the cost of extensive memory. *Parameter-Efficient Transfer Learning* (Guo et al., 2021) also known as *Parameter-Efficient Fine-tuning* (PEFT) (Chen et al., 2023) or *Delta-Tuning* (Ding et al., 2023), offers a solution to this problem. PEFT involves training a minimal subset of parameters, either selected from existing ones or newly added (Lialin et al., 2023). This technique notably reduces memory and storage needs, as only the modified parameters need to be tuned during training and stored post-training. Various mechanisms are employed in PEFT: **(i) Adapter:** One prominent PEFT technique is 'Adapter' training (Houlsby et al., 2019), involving the integration of a bottleneck feed-forward network at each transformer block. **(ii) LoRA:** Another PEFT method, LoRA (Hu et al., 2022), is developed to identify a low-rank delta within specific parameter matrices. **(iii) Soft Prompt Tuning** Lester et al. (2021) is a further PEFT technique that concatenates a trainable matrix to the input embeddings. The columns of this trainable matrix are referred to as *soft prompts*. Although not the leading technique in terms of performance among other PEFT techniques, soft prompt tuning

is renowned for its exceptional parameter efficiency. *Soft Prompt Tuning* is also the central focus of this paper. Different strategies are proposed for an efficient soft prompt tuning:

**(i) Prompt layers reparameterization:** *Residual Prompt Tuning* (Razdaibiedina et al., 2023) is an example of reparameterization of prompt layers employing residual reparameterization to stabilize the prompt tuning process. It uses a randomly initialized autoencoder connected with a residual link.

**(ii) Pre-trained prompts as initial states:** another strategy involves using pre-trained prompts as initial states for new prompts. An example is Soft Prompt Transfer (SPoT) (Vu et al., 2022), which trains a prompt on one or more source tasks and then utilizes it to initialize the prompt for a target task. The selection of appropriate source tasks is crucial in this approach, and a retrieval algorithm is employed to identify similar tasks in a semantic task space.

**(iii) Combined approach:** approaches like Intrinsic Prompt Tuning (IPT) (Qin et al., 2021), AT-TEMPT (Asai et al., 2022), PANDA (Zhong et al., 2022), or MPT (Wang et al., 2023) combine usage of both reparameterization and pre-trained soft prompts. IPT decomposes the pre-trained soft prompts of diverse NLP tasks into a shared low-dimensional subspace by training an autoencoder. Subsequently, the decoder part of the autoencoder is utilized to facilitate learning new prompts in reduced dimensions. ATTEMPT trains an attention layer to combine the right pre-trained prompts using softmax. PANDA uses a knowledge distillation technique to transfer the "knowledge" from the source prompt to the target prompt. MPT trains a single transferable prompt by distilling knowledge from multiple task-specific source prompts.

The training of soft prompts presents notable challenges as highlighted in several studies (Qin et al., 2021; Li & Liang, 2021); particularly, (i) fine-tuning soft prompts is optimization-intensive, particularly with limited data and smaller model sizes in T5 family between 50 to 300 million parameters (Lester et al., 2021); (ii) although typically trainable, soft prompts converge considerably slower compared to full fine-tuning and other delta-tuning methods (Ding et al., 2022). These issues constitute the primary focus of our work.

The contributions of our work can be summarized in two folds: **(i)** we propose SUPERPOS-PROMPT, an innovative reparameterization technique that formulates prompts as superpositions on multiple token embeddings. These token embeddings are sampled vectors from the embedding layer of the language model. This approach enables enhanced stability in prompt tuning using diverse information emanating from multiple token embeddings. This strategy facilitates the learning of a new task representation utilizing a combination of multiple task embeddings. We show that SUPERPOS-PROMPT approach almost consistently outperforms existing relevant soft prompt tuning approaches in 13 Glue and SuperGlue benchmarking tasks. **(ii)** Our research indicates that omitting dropout (Srivastava et al., 2014) from the original network can yield more efficient and expedited convergence in prompt tuning. To the best of our knowledge, this observation has not been addressed in prior studies.

## 2 BACKGROUND

**Full Fine-tuning** involves starting with pre-trained weights and then adjusting all of these weights based on the training data of the new tasks. For example, if we have a new classification dataset $\mathbb{T}$ and the weights of our model, written as $\theta$, we aim to miximize the log likelihood using pre-trained weights as our starting point.

$$\max_{\theta} \sum_{\boldsymbol{X}, y \in \mathbb{T}} \log P_{\theta}(y \mid \boldsymbol{X})$$

**Parameter-Efficient Fine-tuning** involves adding new weights or tune only subset of original weights without changing the other parameters $\theta$. if we denote $\theta'$ as our new parameters it means:

$$\max_{\theta'} \sum_{\boldsymbol{X}, y \in \mathbb{T}} \log P_{\theta}(y \mid \boldsymbol{X}; \theta')$$

**Prompt tuning** is a type of Parameter-Efficient Fine-tuning (PEFT) method where new weights are added only to the model's input by concatenation, without altering $\theta$. In simpler terms, it implies

that we search only in the parameter space $\boldsymbol{P}$ to optimize our model:

$$\max_{\boldsymbol{P}} \sum_{\boldsymbol{X}, y \in \mathbb{T}} \log P_{\theta}(y \mid [\boldsymbol{P}|\boldsymbol{X}])$$

To explain further, if we have a sequence of $l$ tokens, like $\{x_1, x_2, ..., x_l\}$, the model first turns the tokens into a matrix $\boldsymbol{X} \in \mathbb{R}^{e \times l}$, where $l$ is the number of input tokens and $e$ is the dimension of the embedding space. The goal is to find the best soft prompts for our task. These soft prompts are written as $\boldsymbol{P} \in \mathbb{R}^{e \times n}$, where $n$ is the number of the soft prompts. The model then takes the joined matrix $[\boldsymbol{P}|\boldsymbol{X}] \in \mathbb{R}^{e \times (n+l)}$ as input (Lester et al., 2021). This is illustrated in Figure 1.(a).

## 3 Approach

Our objective is to enhance the model's ability to learn and refine soft prompts effectively utilizing multiple token embeddings. This technique is grounded in the observation that initiating the prompt with token representations is generally more beneficial compared to beginning with random vectors (Lester et al., 2021). However, a question arises: how can we employ more than one token embedding for each prompt embedding? We address this issue by adopting a superposition—a weighted sum of several chosen tokens for each prompt embedding, as illustrated in Figure 1.(b).

**SuperPos-Prompt:** We start by randomly selecting $m$ unique token embeddings from the token embedding layer, denoted as $\boldsymbol{e}_1, \boldsymbol{e}_2, ..., \boldsymbol{e}_m$. These are organized as columns of the matrix $\boldsymbol{E} \in \mathbb{R}^{e \times m}$. To compute each prompt token $\boldsymbol{p}_i$, this matrix is multiplied by a vector $\boldsymbol{p}'_i \in \mathbb{R}^m$. During our tuning process, both the matrix $\boldsymbol{E}$ and each $\boldsymbol{p}'_i$ are jointly optimized.

$$\forall i \in \{1, 2, \ldots, n\} \quad \boldsymbol{p}_i = \boldsymbol{E}\boldsymbol{p}'_i = \begin{bmatrix} | & | & & | \\ \boldsymbol{e}_1 & \boldsymbol{e}_2 & \cdots & \boldsymbol{e}_m \\ | & | & & | \end{bmatrix} \begin{bmatrix} | \\ \boldsymbol{p}'_i \\ | \end{bmatrix} = \sum_{j=1}^{m} p'_{ij} \boldsymbol{e}_j$$

During our experiments, we noticed a problem where the inclusion of weight decay in the optimizer led to a reduction in the norm of $\boldsymbol{E}$, resulting in significant information loss in this layer. To combat this, we reparameterize the matrix $\boldsymbol{E}$ as the sum of two matrices: $\boldsymbol{E}_{\text{freeze}}$ and $\Delta\boldsymbol{E}$. In this arrangement, only $\Delta\boldsymbol{E}$ is adjusted while $\boldsymbol{E}_{\text{freeze}}$ remains constant. This strategy effectively counters the negative impact of weight decay on the original embeddings, allowing the model to learn a $\Delta\boldsymbol{E}$ with a lower norm and thus minimally altering the embeddings. For initialization, the matrix $\Delta\boldsymbol{E}$ is set as a zero matrix.

$$\boldsymbol{E} = \boldsymbol{E}_{\text{freeze}} + \Delta\boldsymbol{E} \qquad \Delta\boldsymbol{E}_{\text{init}} = 0_{e \times m}$$

In our experiments, we employed identical initial token embeddings for each prompt while permitting each to adapt uniquely, yielding independent $\Delta\boldsymbol{E}_i$ for every prompt. The final formula to compute each prompt $\boldsymbol{p}_i$ is delineated below and the illustration is provided in Figure 1.(f):

$$\boldsymbol{p}_i = (\boldsymbol{E}_{\text{freeze}} + \Delta\boldsymbol{E}_i)\boldsymbol{p}'_i$$

COMPARISON TO SIMILAR PROMPT TUNING APPROACHES

**Intrinsic Prompt Tuning (IPT)** (Qin et al., 2021) involves training an autoencoder during the *Multi-task Subspace Finding* phase. Post this phase, the decoder part of the autoencoder is employed in the training of new prompts, a stage referred to as *Intrinsic Subspace Tuning* (Figure 1.(d)). In contrast, our approach, SUPERPOS-PROMPT, sidesteps this complexity. We construct the decoder layer by utilizing token embeddings selected directly from the embedding layer. This step negates the need for pre-trained soft prompts and the associated training of an autoencoder, as illustrated in Figure 1.(e).

**ATTEMPT** (Asai et al., 2022) also has similarities with our method, but it relies on pretrained source prompts instead of token embeddings, and employs softmax weighting instead of superposition. Through our experiments, we noticed that utilizing superposition is more efficient than softmax weighting as we showed in §A.2.

**Residual Prompt Tuning:** Our approach shares similarities with *Residual Prompt Tuning* (Razdaibiedina et al., 2023), as both employ reparameterization to achieve improved and more rapid

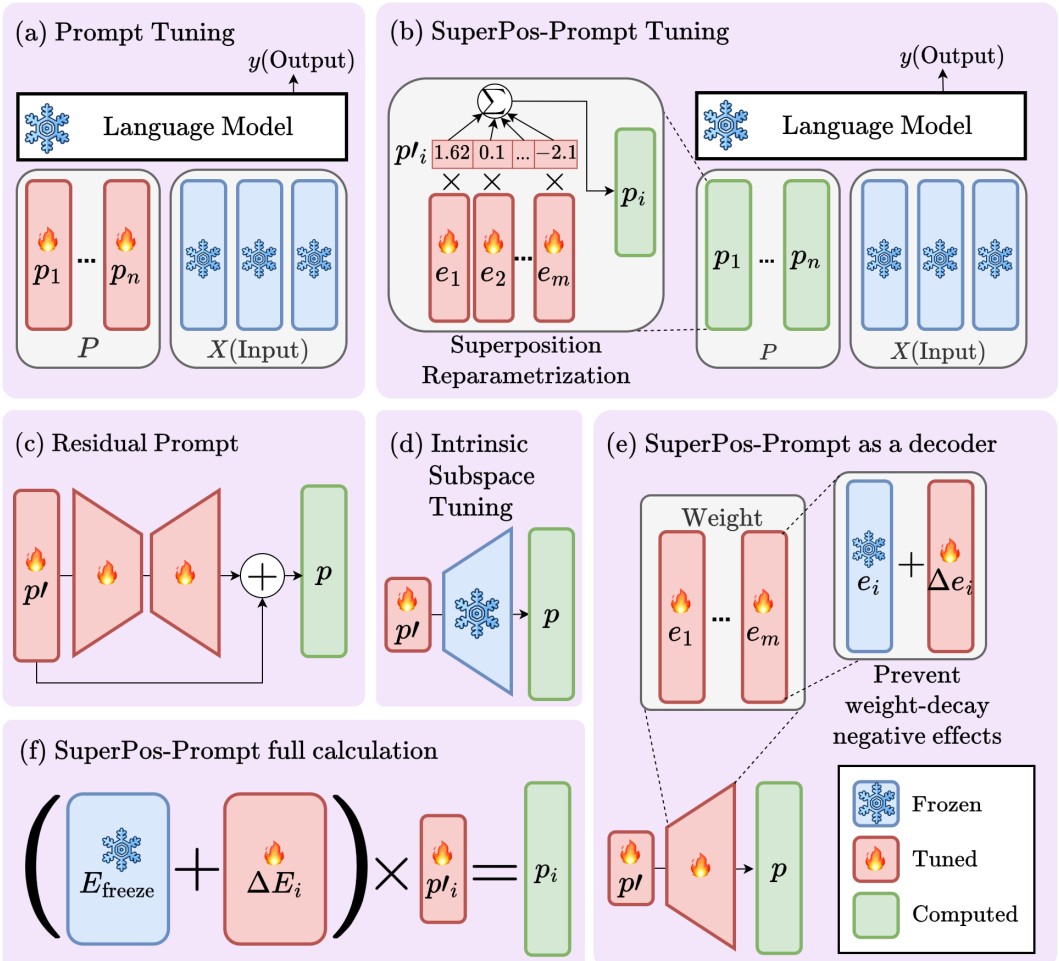

Figure 1: Overview of different prompt tuning methods: **(a.)** *Simple Prompt Tuning:* This method adjusts the prompt embeddings, $P$, which are then concatenated with the input embeddings. **(b.)** SUPERPOS-PROMPT Tuning: Employs a mixture of embeddings as a weighted sum, $e_j; 1 \le j \le m$, based on their weight in $p'_i$. All $e_j$s and vector $p'_i$ are co-tuned. **(c.)** *Residual Prompt Tuning:* Utilizes an autoencoder with residual connection reparametrization. **(d.)** *Intrinsic Subspace Tuning:* Employs a pre-trained decoder to map lower-dimension prompts to the model's dimension. **(e.)** SUPERPOS-PROMPT can also be interpreted as a linear up-projection initialized with sampled embeddings. **(f.)** SUPERPOS-PROMPT Tuning full calculation consist of an addition to prevent weight-decay negative effects and matrix multiplication to calculate superposition of embeddings.

convergence, avoiding the use of pretrained soft prompts. However, *Residual Prompt Tuning* utilizes an encoder-decoder model with a residual connection and is tuned end-to-end, as shown in Figure 1.(c). In contrast, our model is simpler, having only half the components to tune. It consists only of an up-projection layer, and by using pretrained token embeddings to initialize the decoder's weights, it offers a more advantageous starting point.

We evaluate our method against vanilla prompt tuning (Lester et al., 2021), residual prompt tuning (Razdaibiedina et al., 2023), and *ATTEMPT* (Asai et al., 2022). We intentionally excluded *IPT* (Qin et al., 2021) from our comparison. The exclusion is due to *IPT*'s requirement for 100 pre-trained source prompts to train an auto-encoder. Since they utilize BART (Lewis et al., 2020) as their backbone model, their autoencoder was incompatible with our framework. Training a new auto-encoder was not feasible as we lacked access to the necessary 100 pre-trained source prompts.

# 4 EXPERIMENTS

## 4.1 DATASET

In previous studies, smaller datasets have presented substantial challenges for prompt tuning techniques (Ding et al., 2022). To effectively contrast various methods, we have selected several tasks/datasets from both GLUE (Wang et al., 2019b) and SuperGLUE (Wang et al., 2019a), comprising both small and large datasets. The datasets employed in our study are the Quora Question Pairs (QQP) (DataCanary et al., 2017), Question NLI (QNLI), MultiNLI (MNLI) (Williams et al., 2018), The Stanford Sentiment Treebank (SST-2) (Socher et al., 2013), Semantic Textual Similarity Benchmark (STS-B) (Cer et al., 2017), Microsoft Research Paraphrase Corpus (MRPC) (Dolan & Brockett, 2005), The Corpus of Linguistic Acceptability (CoLA) (Warstadt et al., 2019), Multi-Sentence Reading Comprehension (MultiRC) (Khashabi et al., 2018), Recognizing Textual Entailment (RTE), CommitmentBank (CB), Choice Of Plausible Alternatives (COPA) (Gordon et al., 2012), Words in Context (WiC) (Pilehvar & Camacho-Collados, 2019), and BoolQ (Clark et al., 2019).

## 4.2 BASE LANGUAGE MODEL

In this study, we employ the T5 model family for conducting experiments (Raffel et al., 2020). Our approach to the classification task involves conditional generation, wherein the output comprises a string of tokens, each symbolizing a class label. This study exclusively modifies the encoder segment of the T5 model by integrating soft prompts. Given the constraints of computational resources, our analysis is confined to the small and base model sizes. Specifically, we deploy two LM-adapted versions of T5v1.1, namely *t5-small-lm-adapt* and *t5-base-lm-adapt* (Lester et al., 2021).

Previous research, including studies such as the Residual Prompt and ATTEMPT, have highlighted concerns regarding the stability and tuning difficulties of T5v1.1-LM adapt when used as a backbone for prompt tuning tasks (Razdaibiedina et al., 2023; Asai et al., 2022). These studies eventually switched to the original T5 checkpoint. However, utilizing the pretrained T5 original checkpoint raises concerns. Since this checkpoint is already trained on the GLUE and SuperGLUE datasets, the model does not need to learn a new task, only requiring the appropriate prompt to utilize previously acquired knowledge (Raffel et al., 2020). This situation may produce misleading results, obscuring the true performance and meaningfulness of the ultimate comparison. Therefore we implemented and tested their methods using the provided hyperparemeters on T5v1.1-LM adapt.

## 4.3 ABLATION STUDY

In SuperPos prompt tuning, a key hyperparameter is the number of tokens sampled for superposition, denoted as $m$. Figure 2.(c) shows the impact of different $m$ values on the performance of SUPERPOS-PROMPT across various tasks. On the x-axis, we display the number of tokens ($m$), and the y-axis shows the highest performance score achieved. We observe that an increase in the number of sampled tokens generally leads to better results, but improvements tend to level off after reaching 128 tokens. Based on this finding, we set the number of sampled tokens in our method to 128.

## 4.4 EXPERIMENT SETUP

For our experiments, the following configurations were employed:

**All of Prompt Tuning Methods:** We appended 10 prompt tokens to the input. Each method was tested under two conditions: with and without dropout, running for a total of 80 epochs. No learning rate scheduler was used, and the AdamW optimizer (Loshchilov & Hutter, 2019) was employed.

**Simple Prompt Tuning:** Prompts were initialized by sampling 10 unique token embeddings from the embedding layer, using a learning rate of 0.01 and a weight decay of 0.01.

**Residual Prompt Tuning:** Prompts were initialized by sampling 10 unique token embeddings from the embedding layer, with a learning rate of 0.3 and a weight decay of 0.01, as specified in the original paper (Razdaibiedina et al., 2023), we set the bottleneck size to 128 to be comparable to our method.

Table 1: Results on some tasks from GLUE and SuperGLUE dataset set with 10-token prompts and training for 80 epochs. For tasks with two metrics, the average score is reported. Numbers marked with † means that T5 model doesn't converge to always generate valid labels. So the score will be zero. The full fine-tuning are reported as a comparsion baseline.

| Task→ Method↓ | Dropout | GLUE | | | | | | | SuperGLUE | | | | | | Avg. |
| | | QQP | QNLI | MNLI | SST-2 | STS-B | MRPC | CoLA | MultiRC | RTE | CB | COPA | WiC | BoolQ | |
| | | F1/Acc. | Acc. | Acc. | Acc. | PCC/ρ | F1/Acc. | MCC | F1a/EM | Acc. | F1/Acc. | Acc. | Acc. | Acc. | - |
| **T5v1.1 Small LM-Adapted** | | | | | | | | | | | | | | | |
| Simple PT | ✓ | 58.2/65.5 | 50.6 | 33.2 | 79.4 | 9.8/7.9 | 81.2/68.4 | 0.0 † | 17.3/0.3 | 52.3 | 0.0/0.0 † | 0.0 † | 50.6 | 62.2 | 37.1 |
| Simple PT | ✗ | 70.8/75.3 | 72.8 | 50.7 | 84.9 | 0.0/0.0 † | 82.5/71.3 | 0.0 † | 22.6/0.6 | 49.1 | 0.0/0.0 † | 0.0 † | 57.4 | 62.6 | 41.5 |
| ATTEMPT | ✓ | - | - | - | - | 0.0/0.0 † | 0.0/0.0 † | 0.0 † | 0.0/0.0 † | 52.0 | 0.0/0.0 † | 58.0 | 0.0 † | 0.0 † | - |
| ATTEMPT | ✗ | - | - | - | - | 83.3/83.2 | 0.0/0.0 † | 0.0 † | 0.0/0.0 † | 59.9 | 0.0/0.0 † | 57.0 | 64.3 | 0.0 † | - |
| Residual PT | ✓ | 70.6/74.9 | 61.8 | 34.6 | 82.8 | 69.7/72.4 | 81.9/71.1 | 0.5 | 59.9/0.8 | 52.7 | 49.6/71.4 | 56.0 | 52.4 | 62.3 | 54.9 |
| Residual PT | ✗ | 73.3/78.2 | 79.2 | 60.7 | 85.1 | 80.8/80.6 | 88.3/83.3 | 20.6 | 59.8/4.4 | 59.6 | 68.6/73.2 | 56.0 | 58.2 | 64.7 | 63.8 |
| SuperPos PT | ✓ | 74.4/79.9 | 82.9 | 66.7 | 88.8 | 82.9/82.8 | 88.4/82.6 | 23.4 | 59.9/0.8 | 58.5 | 39.6/60.7 | 56.0 | 58.6 | 62.4 | 63.3 |
| SuperPos PT | ✗ | **79.1/83.3** | **85.3** | **71.7** | **89.8** | **84.0/84.0** | **89.9/85.8** | **38.9** | **66.6/16.7** | **64.6** | **73.6/76.8** | 58.0 | **65.7** | **68.9** | **70.2** |
| Full Fine-tuning | ✓ | 87.4/90.5 | 89.5 | 82.9 | 92.1 | 85.8/85.5 | 89.6/84.8 | 42.0 | 68.5/19.3 | 66.1 | 47.9/69.6 | 57.0 | 66.5 | 71.1 | 71.7 |
| **T5v1.1 Base LM-Adapted** | | | | | | | | | | | | | | | |
| Simple PT | ✓ | 54.3/38.2 | 50.5 | 34.8 | 85.0 | 0.0/0.0 † | 81.2/68.4 | 0.0 † | 2.5/0.3 | 53.1 | 0.0/0.0 † | 0.0 † | 50.6 | 62.6 | 35.3 |
| Simple PT | ✗ | 0.0/0.0 † | 76.9 | 0.0 † | 92.2 | 0.0/0.0 † | 82.0/70.6 | 24.8 | 55.6/2.1 | 53.4 | 0.0/0.0 † | 59.0 | 57.7 | 0.0 † | 36.1 |
| ATTEMPT | ✓ | - | - | - | - | 0.0/0.0 † | 0.0/0.0 † | 44.6 | 0.0/0.0 † | 56.0 | 0.0/0.0 † | 55.0 | 0.0 † | 0.0 † | - |
| ATTEMPT | ✗ | - | - | - | - | 0.0/0.0 † | 0.0/0.0 † | 53.7 | 67.5/17.8 | 56.0 | 0.0/0.0 † | 0.0 † | **69.0** | 70.1 | - |
| Residual PT | ✓ | 72.1/75.0 | 58.0 | 34.8 | 91.3 | 81.6/81.7 | 82.0/70.3 | 0.0 † | 59.9/0.8 | 52.7 | 43.6/64.3 | 58.0 | 54.2 | 62.8 | 56.0 |
| Residual PT | ✗ | 76.1/81.4 | 83.3 | 70.7 | 92.7 | 86.2/86.1 | 87.4/82.8 | 44.7 | 63.9/11.3 | 70.0 | **82.6/80.4** | 60.0 | 64.3 | 65.3 | 70.8 |
| SuperPos PT | ✓ | 79.0/83.1 | 79.2 | 76.5 | 94.0 | 86.2/86.6 | 89.1/83.6 | 45.4 | 68.7/18.2 | 57.4 | 44.8/66.1 | 58.0 | 58.3 | 62.3 | 68.0 |
| SuperPos PT | ✗ | **81.9/86.3** | **89.8** | **81.0** | **94.2** | **88.6/88.5** | **89.7/85.5** | **56.5** | **72.9/24.9** | **70.4** | 78.3/82.1 | **62.0** | 67.6 | **74.0** | **75.8** |
| Full Fine-tuning | ✓ | 88.3/91.1 | 92.7 | 88.1 | 94.8 | 90.1/89.8 | 91.9/88.2 | 53.0 | 76.2/35.3 | 72.9 | 53.5/76.8 | 57.0 | 69.3 | 78.9 | 76.7 |

**ATTEMPT (Asai et al., 2022):** $P_{target}$ prompts were initialized by sampling ten unique token embeddings from the embedding layer. To avoid leakage between training and testing data, we excluded QQP, QNLI, MNLI, SST-2 datasets from the evaluation, as these task pretrained prompts were used during the training of new prompts. To align with the hyperparameters from the original ATTEMPT paper, the learning rate is set to 0.3, with a weight decay of 0.00001, and a bottleneck size of $\mathcal{G}$ set to 100.

**SuperPos Prompt Tuning:** Prompts in superposition were initialized with 128 unique token embeddings, shared across all 10 prompt tokens. The learning rate was 0.01 with a weight decay of 0.00001.

**Full Fine-tuning:** We opted for a lower learning rate of 0.00001 to preserve the original weights more effectively.

## 5 RESULTS

Our experimental results are compiled in Table 1. Runs generating invalid labels, a possible consequence of conditional generation, are denoted with † and scored as 0. Standard metrics from the GLUE and SuperGLUE benchmarks are used for each task.

**Impact of Dropout:** As shown in Figure 2.(a) and Table 1 eliminating dropout from the frozen model enhanced not only the performance of the model but also accelerated convergence. This trend was also evident in experiments with *Residual Prompt*, *ATTEMPT*, and SUPERPOS-PROMPT tuning methods. We hypothesize that dropout, being a form of regularization to prevent overfitting, may excessively constrain prompt tuning. Since tuning only 10 prompts inherently limits flexibility, additional dropout may lead to underperformance.

**SuperPos-Prompt Performance:** According to Table 1, SUPERPOS-PROMPT excelled over *Residual Prompt* tuning, showing a significant average score increase of +6.4 in *T5v1.1-Small* and +5 in *T5v1.1-Base*. Our method has superior performance on most tasks that *ATTEMPT* were tested on. In some cases, it even surpassed full fine-tuning methods. A more detailed comparison of some selected tasks learning curves, based on *T5v1.1 Base LM-Adapted* experiments, is available in Figure 2.(b). Among the compared methods, SUPERPOS-PROMPT generally achieved better performance

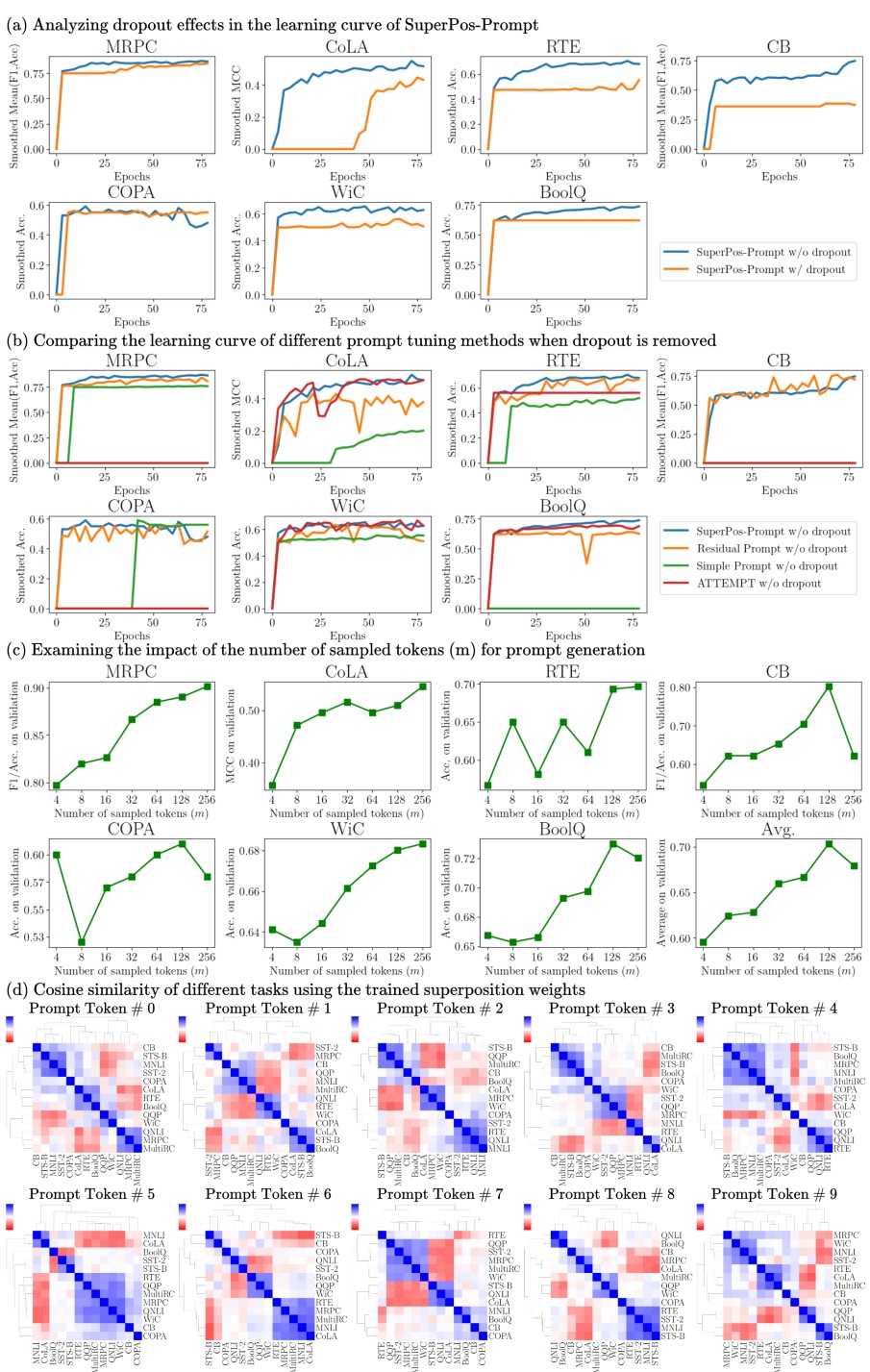

Figure 2: This figure illustrates results from our experiment using 'T5v1.1 Base LM-Adapted' as the foundation. **(a)** Learning curves comparing dropout effects on SuperPos-Prompt for selected tasks. **(b)** Learning curves comparing various prompt tuning methods across selected tasks, conducted without dropout. **(c)** Ablation study on the effect of sampled token count ($m$) for SuperPos-Prompt, with the x-axis representing sample token count and the y-axis indicating peak performance for the relevant metric. **(d)** Analysis of cosine similarity in superposition weights for each prompt token across all tasks.

Table 2: Mean and standard deviation of standardized overall scoring across thirteen different tasks. This table facilitates a comparison of method stability, where a lower standard deviation indicates higher stability across tasks. Note: ATTEMPT results are excluded as it was not evaluated on four tasks from thirteen tasks.

| Method↓ | Dropout | T5v1.1 Small LM-Adapted | T5v1.1 Base LM-Adapted |
|---------|---------|-------------------------|------------------------|
| Simple PT | ✓ | 17.1±26.4 | 17.2±25.2 |
| Simple PT | ✗ | 28.9±29.5 | 30.8±32.6 |
| Residual PT | ✓ | 44.7±31.3 | 49.5±32.8 |
| Residual PT | ✗ | 65.9±20.0 | 83.2±10.2 |
| SuperPos PT | ✓ | 66.9±17.8 | 75.9±18.5 |
| SuperPos PT | ✗ | 81.7±**9.7** | 93.6±**4.7** |
| Full Fine-tuning | ✓ | 85.2±9.0 | 97.4±5.7 |

and faster convergence. All learning curves are without dropout variant of that methods as most of the time this variant reached their best performances, as detailed inTable 1.

**Other Prompt Tuning Methods Performances:** The performance of *Residual Prompt* and *ATTEMPT* did not meet the levels reported in their respective papers. This discrepancy may stem from their use of T5 checkpoints trained specifically on these tasks. Unable to replicate their results, we tested our method using identical checkpoint and found it surpassed their reported numbers. For more details, see §A.1.

**Stability Analysis:** To compare the stability of various methods, we normalized and scaled the performance of each task across these methods. This process, referred to as "standardized overall scoring", is described by Yu et al. (2023) and is employed in evaluating Large Language Models (LLMs). To determine stability, we calculated the mean and standard deviation of these scores for each method over thirteen tasks. A method demonstrating a lower standard deviation suggests greater stability, indicating consistent performance across various tasks. As shown in Table 2, our method has a standard deviation half that of the RESIDUAL PROMPT, thus exhibiting superior stability in prompt tuning tasks, closely rivaling stability of full fine-tuning.

**Analysis on Learned SuperPos-Prompt:** We performed a cosine similarity analysis on the learned superposition weights ($p_i'$) for each prompt across different tasks. The resulting similarity matrices are presented in Figure 2.(d). Each prompt's token similarity matrix reveals distinct patterns, suggesting unique task-specific encodings. However, we found no clear correlation between these patterns and the task descriptions. Notably, tasks with limited data and fewer training steps, such as CB, COPA, and RTE, tend to have the most distinctive prompts.

## 6 CONCLUSIONS

In this work, we made two primary contributions that enhance the field of prompt tuning for language models, especially when fine-tuning datasets are small and existing soft prompt tuning approaches fall short.

First, we observed a notable improvement in the efficiency and speed of convergence in prompt tuning upon excluding dropout from the frozen network. This observation, which has not been explored in existing literature, holds consistently across most scenarios, enhancing the performance of RESIDUAL PROMPT, ATTEMPT, and SUPERPOS-PROMPT tuning methods. Our findings underscore the importance of continually reassessing established network parameters and practices to unearth potential enhancements.

Our second key contribution was the introduction of SUPERPOS-PROMPT, a novel reparameterization technique for soft prompt tuning. This method, leveraging the superpositions of sampled pretrained token embeddings, enhances stability in prompt tuning and obviates the need for pretrained source prompts. SUPERPOS-PROMPT consistently outperformed *Residual Prompt* tuning,

showcasing an average score increase of $+6.4$ in *T5-Small* and $+5.0$ in *T5-Base* across all thirteen GLUE and SuperGLUE benchmarks used in this study. Remarkably, SUPERPOS-PROMPT not only exceeded the performance of *Residual Prompt* tuning but also, in certain instances, showed superior performance to the full fine-tuning approach. Additionally, we observed a clear correlation between the number of sampled tokens on SUPERPOS-PROMPT and performance scores, with an optimal plateau at 128 tokens.

Looking forward, the exploration of integrating pre-trained source prompts stands as a promising avenue for further enhancing model performances. We anticipate that our work will spur innovative and more efficient uses of pre-trained source prompts in the future, reinforcing the importance of this research in the ever-evolving field of language model tuning and optimization. Future work includes a more extensive comparison of SUPERPOS-PROMPT with a broader range of prompting techniques in different dataset scenarios, an endeavor constrained in this study by computational resource limitations. Additionally, while this study exclusively explored language models, we anticipate the extension of this approach to additional foundation models across various modalities, as well as multimodal foundation models.

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

# A APPENDIX

Table 3: This table presents additional results and comparisons, including those from the Super-Pos prompt method trained on the T5 Base checkpoint. Results from methods marked with ⋆ are sourced from their respective papers (Asai et al., 2022; Razdaibiedina et al., 2023). It also shows the impact of softmax application and GPT-3.5-Turbo's one-shot performance across various datasets. Unreported values are indicated by '-'. In the residual prompt tuning study, tasks with two metrics are reported as an average score, not separately.

| Task→ | # Prompts | Softmax | Dropout | GLUE | | | | | | | SuperGLUE | | | | | | Avg. |
| Method↓ | | | | QQP | QNLI | MNLI | SST-2 | STS-B | MRPC | CoLA | MultiRC | RTE | CB | COPA | WiC | BoolQ | |
| | | | | F1/Acc. | Acc. | Acc. | Acc. | PCC/$\rho$ | F1/Acc. | MCC | F1a/EM | Acc. | F1/Acc. | Acc. | Acc. | Acc. | - |
| T5 Base | | | | | | | | | | | | | | | | | |
| SuperPos PT | 10 | ✗ | ✗ | **87.8/90.8** | **93.5** | **86.0** | **94.4** | **90.2/90.1** | **92.4/89.5** | **59.7** | **77.7/40.9** | **80.1** | **97.4/96.4** | **66.0** | **67.6** | **81.3** | **81.2** |
| ATTEMPT ⋆ | 100 | ✓ | ✓ | -/90.3 | 93.0 | 84.3 | 93.2 | 89.7/- | -/85.7 | 57.4 | 74.4/- | 73.4 | -/78.6 | - | 66.8 | 78.8 | - |
| Residual PT ⋆ | 10 | ✗ | ✓ | - | - | - | - | - | - | - | 59.3 | 70.4 | 79.2 | 58.3 | 66.8 | 77.9 | - |
| T5v1.1 Small LM-Adapted | | | | | | | | | | | | | | | | | |
| SuperPos PT | 10 | ✗ | ✗ | 79.1/83.3 | 85.3 | 71.7 | 89.8 | 84.0/84.0 | 89.9/85.8 | 38.9 | 66.6/16.7 | 64.6 | 73.6/76.8 | 58.0 | 65.7 | 68.9 | **70.2** |
| SuperPos PT | 10 | ✓ | ✗ | 69.6/75.2 | 76.0 | 42.7 | 82.9 | 45.5/43.3 | 82.4/73.0 | 4.6 | 47.5/0.9 | 52.0 | 49.9/71.4 | 57.0 | 56.4 | 62.3 | 54.9 |
| T5v1.1 Base LM-Adapted | | | | | | | | | | | | | | | | | |
| SuperPos PT | 10 | ✗ | ✗ | 81.9/86.3 | 89.8 | 81.0 | 94.2 | 88.6/88.5 | 89.7/85.5 | 56.5 | 72.9/24.9 | 70.4 | 78.3/82.1 | 62.0 | 67.6 | 74.0 | 75.8 |
| GPT-3.5-Turbo | | | | | | | | | | | | | | | | | |
| 1 Shot | | | | 76.3/79.2 | 70.9 | 58.5 | 94.0 | 34.6/34.1 | 84.6/77.0 | 46.1 | 77.9/34.1 | 70.8 | 55.6/62.5 | 95.0 | 58.8 | 69.6 | 67.1 |

## A.1 T5 ORIGINAL CHECKPOINT

As noted earlier, some studies like Residual Prompt and ATTEMPT used the original T5 checkpoint, trained on these tasks, instead of the T5v1.1 LM-Adapted checkpoint. Our replication efforts with the T5v1.1 LM-Adapted checkpoint yielded unsatisfactory results. Consequently, we also adopted the original T5 checkpoint in our method for a fair comparison. As illustrated in Table 3, our approach outperformed results that were reported in these studies. This outcome is significant, especially considering that the ATTEMPT method utilized ten times more prompt tokens and also used pre-trained source prompts for initialization.

## A.2 SOFTMAX EFFECT

In our experiments, we also applied a softmax function to the superposition weights. This approach aligns more closely with an attention mechanism, effectively computing an expected value. The mathematical representation is as follows:

$$\forall i \in \{1, 2, \ldots, n\} \quad \boldsymbol{p}_i = \boldsymbol{E}\, \text{Softmax}(\boldsymbol{p}_i') = \frac{\sum_{j=1}^{m} \exp\left(p_{ij}'\right) \boldsymbol{e}_j}{\sum_{j=1}^{m} \exp\left(p_{ij}'\right)}$$

However, this modification resulted in diminished performance, as indicated in Table 3. Therefore we didn't use softmax in our main experiments.

## A.3 GPT3 FEW-SHOT PERFORMANCE

For comparison purposes, we conducted experiments on these datasets using the *GPT-3.5-turbo* model. The model was evaluated with in-context learning, employing 1-shot examples from each category. The results can be found in Table 3.

