# OpenReview forum: "SuperPos-Prompt: Enhancing Soft Prompt Tuning of Language Models with Superposition of Multi Token Embeddings"
_ICLR.cc/2024/Conference — Submitted to ICLR 2024_

### Official Review · Reviewer_xLGd · 2023-10-30

**Soundness:** 2 fair
**Presentation:** 3 good
**Contribution:** 3 good
**Rating:** 6
**Confidence:** 3

**Summary:**

This paper introduce a novel soft prompt tuning technique, which is a new reparameterization technique that employs the superposition of multiple pre-trained vocabulary embeddings to improve the learning of soft prompt turning, and is able to improve without relying on the pre-trained soft prompts.The experiments tuned the LM-adapted T5 model on a smaller scale dataset. Significant improvement was achieved compared to the Residual Prompt tuning technique.

**Strengths:**

The experimental results presented in this paper are significant and appealing from the point of view of the development of prompt-tuning techniques as well as fine-tuning of small-scale datasets. The authors expand the reparameterization methods in the field of soft prompt tuning techniques and provide a more detailed experimental demonstration. In conclusion, the SUPERPOS-PROMPT presented in this paper is of great value and helps to advance the development of related research. Detailed and sufficient results are provided in the main text. Thus, the SUPERPOS-PROMPT proposed in this paper is of great value and helps to promote the development of research in the field of prompt tuning.

**Weaknesses:**

1. the paper presents experimental results from the innovations without sufficient justification and explanation, e.g., what are the key explainable innovations that contribute to the performance improvement over similar techniques IPT, ATTEMPT, and Residual Prompt Tuning.
2. The validity of large language models needs to be further demonstrated.
3. The validity for LMs beyond T5 is unknown.

**Questions:**

Why does the author only choose T5 as the model for evaluation? Especial when the authors already noticed that T5 has checkpoints that pretrained on GLUE and SuperGLUE datasets. Therefore, I doubt the contribution of proposed method if the method does not lead to a progress on the SOTA results compared to either pretraining or other fine-tuning approaches.

---

> ### Author Response · Authors · 2023-11-23
>
> Dear Reviewer,
>
> We are grateful for your insightful feedback, which has not only been invaluable but has also substantially elevated the quality of our article.  To avoid repeating information about the additional experiments and analyses, we kindly direct your attention to 'Response to Reviewer 4x9F'. This document contains detailed information about the changes in several key areas:
> 1. Expansion of the Evaluations
> 2. Stability Analysis of the Results
> 3. Enhanced Baseline Comparisons
> 4. Additional Experiments with T5-Base
> 5. Visual Analysis of Task Similarity
> 6. Superposition vs. Softmax Analysis
> 7. GPT-3.5-Turbo Inclusion.
>
> In response to the question regarding our choice of the T5 model:
> We selected the T5 model due to its widespread recognition and frequent use in prompt tuning research, as evidenced by its application in foundational studies like the original prompt tuning research and the 'ATTEMPT' paper. Our focus on T5 aligns with prevalent practices in the field and allows for a focused, in-depth analysis within our study's scope. It's important to note that the objective of parameter-efficient fine-tuning, as demonstrated by our method, is not necessarily to surpass state-of-the-art (SOTA) results achieved by full fine-tuning methods. Instead, our aim is to achieve significant improvements with minimal parameter modifications. Our method demonstrates its effectiveness by showing strong results with only 10 tokens tuned. This is a significant achievement in parameter-efficient tuning, offering performance and consistency on par with full fine-tuning method. While exploring a wider range of models would enrich our research, our current resources limited us to T5. We acknowledge the value of testing our method on other models and will consider this for future work.
>
> Here are some examples of previous recent work on T5:
>
> The Power of Scale for Parameter-Efficient Prompt Tuning (Lester et al., EMNLP 2021)
>
> SPoT: Better Frozen Model Adaptation through Soft Prompt Transfer (Vu et al., ACL 2022)
>
> Residual Prompt Tuning: Improving Prompt Tuning with Residual Reparameterization (Razdaibiedina et al., ACL 2023)
>
>
>
> Regarding the weaknesses highlighted:
>
> Clarity on Innovations: In addition to eliminating dropout during training, another key innovation of our method is the use of multiple token embeddings for each prompt token, different from the standard practice in the field that typically employs a single token embedding per prompt token. This approach enables a more complex and nuanced initialization vital for effective learning and better adaptation across a wide range of tasks. While our method may not be complex, our experiments demonstrate its consistent improvement over benchmark datasets, with minimal overall variance compared to other soft-prompt tuning approaches. It also shows competitiveness with the full fine-tuning approach, often considered the upper bound.

---

### Official Review · Reviewer_HvC5 · 2023-10-30

**Soundness:** 2 fair
**Presentation:** 2 fair
**Contribution:** 2 fair
**Rating:** 3
**Confidence:** 4

**Summary:**

This paper tries to propose a different soft-prompt tuning method. The limited experiments show that the proposed method outperforms residual prompt tuning and original soft prompt tuning.

**Strengths:**

The method’s performance seems better than the compared baselines.

**Weaknesses:**

1. Unclear presentation. For example, if you finally decided to freeze E_{freeze}, how did you initialize it? Is it just randomly initialized or intialized with some pretrained token embeddings?
2. Missing baselines. For example, the authors should at least compare the proposed method with the similar methods they mentioned themselves, such as IPT and ATTEMPT, in a fair setting. “Through our experiments, we noticed that utilizing superposition is more efficient than softmax weighting.” Where is this experiment?
3. Too many ablations are missing so that I find it very hard to understand why the proposed method is so much better than residual prompt tuning. The authors mention 3 variants in the main figure (Figure 1, bef) but have shown no results regarding such variants. Where does the performance improvement come from? Is it mainly from freezing E_{freeze}, or removing softmax, or reducing parameters?

**Questions:**

See weakness

---

> ### Author Response · Authors · 2023-11-23
>
> Dear Reviewer,
>
> We sincerely appreciate your feedback. To streamline the revised version and avoid repeating information about the additional experiments and analyses, we kindly direct your attention to 'Response to Reviewer 4x9F'. This document contains detailed information about the changes in several key areas:
> 1. Expansion of the Evaluations
> 2. Stability Analysis of the Results
> 3. Enhanced Baseline Comparisons
> 4. Additional Experiments with T5-Base
> 5. Visual Analysis of Task Similarity
> 6. Superposition vs. Softmax Analysis
> 7. GPT-3.5-Turbo Inclusion.
>
> Regarding specific concerns:
>
> **1.** Regarding the initialization of $\( E_{\text{freeze}} \)$. In our method, $\( E_{\text{freeze}} \)$ is initialized by selecting $\( m \)$ random unique token embeddings from the language model's embedding layer. We believe the key factor in our method's success and robustness is its ability to utilize more than one token embedding for each prompt embedding. The purpose of $\( \Delta E \)$ in our methodology is to accumulate gradients during training, preserving the norm of the original $\( E \)$ matrix from being affected by weight decay. This reparameterization technique is utilized to maintain the effectiveness of $\( E_{\text{freeze}} \)$ without directly modifying it during the optimization process. We hope this clarification addresses the raised concern.
>
> **2.** In line with your suggestion, we have included ATTEMPT in our set of baselines, enabling a more comprehensive comparison. We also tested our method using T5-Base and compared our results with their reported number in the appendix (Table A3). Regarding the comparison with IPT, we encountered a methodological limitation. The IPT approach utilizes the BART model, whereas our work is based on the T5 model. To conduct a fair comparison, we would need access to a similar number of pretrained source prompts (around 100) to train an equivalent autoencoder. Unfortunately, our resources did not extend to this number of source prompts, and training an autoencoder with a small amount of data would not provide optimal results. We have explained this limitation in our revised manuscript for transparency. Furthermore, as discussed, we have added the experiment comparing superposition efficiency with softmax weighting in the appendix. This experiment provides empirical evidence supporting our claim about the efficiency of superposition over softmax weighting in our context.
>
>
> **3.** We are thankful about the feedback on Figure 1 and the effectiveness of our proposed method compared to residual prompt tuning. To clarify, Figure 1 (b, e, and f) illustrates different aspects of the same method, rather than new variants. Specifically, (b) and (e) offer different visual perspectives of our method, while (f) details its linear algebra computation. Regarding the performance improvement of SuperPos-Prompt, the key enhancement comes from our novel initialization technique, which utilizes more information from the embedding layer. In addition, we have found that omitting dropout also contributes positively to generalization and convergence when prompt tuning the T5 model. These benefits are supported by empirical data presented in Table 1 and Figure 2a, where we compare the outcomes of various prompt tuning methods with and without this modification.

---

### Official Review · Reviewer_gZe6 · 2023-10-31

**Soundness:** 2 fair
**Presentation:** 2 fair
**Contribution:** 3 good
**Rating:** 5
**Confidence:** 4

**Summary:**

This work focuses on soft prompt tuning, a method to efficiently fine-tune pretrained language models with minimal parameter updates (PEFT methods). Soft prompts are challenging to optimize, especially with small datasets. This is empirically observed. PEFT methods such as LoRA and adapters are more popular than soft prompt tuning, prefix tuning etc. because of this reason. When finetuning an LLM on small datasets they seem to suffer from a notable performance drop compared to vanilla as observed in previous works [1]. The study contributes in two ways:
1. Introduction of SUPERPOS-PROMPT: This is a simple to use reparameterization technique that improves the learning of soft prompts. It does so by taking a linear combination of multiple pretrained vocabulary embeddings. The authors conducted experiments across various GLUE and SuperGLUE benchmarks, showing that SUPERPOS-PROMPT outperforms Residual Prompt tuning. It yields an average score improvement of +4.7 on T5-Small and +3.9 on T5-Base, along with faster convergence. SUPERPOS-PROMPT sometimes even outperforms full fine-tuning methods however I have a few questions regarding this result.
2. Omission of Dropout from Frozen Network: The authors demonstrate that excluding dropout from the frozen (unchanged) parts of the network leads to enhanced performance and faster convergence. This improvement is consistent across different scenarios and tuning methods.

Crucially, the proposed approach does not depend on having a proficient pretrained source prompt for initialization, providing significant flexibility and more effective combination of related prompt candidates. The authors argue that this is a common limitation in existing strategies, making the approach more versatile and applicable to a broader range of tasks. However, I believe comparisons with these methods would make the method more insightful.

[1] Hu, Edward J., et al. "Lora: Low-rank adaptation of large language models." arXiv preprint arXiv:2106.09685 (2021).

**Strengths:**

Strengths
1) I agree with the authors that the challenge of soft prompt tuning on small datasets has to be addressed. Although this technique substantially reduces the number of trainable parameters, its diminished performance and slower convergence on smaller datasets makes its use less appealing. Effectively tackling these issues would constitute a significant contribution. Authors make an attempt at addressing these issues.
2) The authors demonstrate that excluding dropout from the frozen (unchanged) parts of the network leads to enhanced performance and faster convergence. This improvement is consistent across different scenarios and tuning methods.
3) I thank the authors for pointing this inference, “Through our experiments, we noticed that utilizing superposition is more efficient than softmax weighting.” This result has more implication since it says linear combination is more beneficial than convex combination for prompt superposition.
4) The authors propose a simple yet effective strategy (empirically) of linear combination of multiple tokens as prompt initialization.

**Weaknesses:**

Thank you for your work. I hope my following suggestions would make the work more robust.

1) The writing requires improvement.
For instance, (a) the authors introduce various methods under three categories: 1) Prompt layers reparameterization, 2) Pre-trained prompts as initial states, and 3) Combined approach. Presenting this multitude of works in the introduction might overwhelm and confuse the reader. As an alternative, the authors could cite one representative work for each category and discuss the limitations of these works in the introduction. Additionally, a dedicated ‘Related Works’ section could be included, where the authors provide a more comprehensive overview of the numerous cited works, allowing readers to delve into further details. This structure would enhance the paper’s readability. (b) Another example is, authors propose “We begin with selecting a linear combination of m unique token embeddings sampled from the token embedding layer, denoted as e1 , e2 , ..., em .” However, they did not mention how to select those ‘m’. Are they randomly sampled?
(c) “Impact of Dropout, Impact of SuperPos-Prompt, Effect of the Number of Sampled Tokens” in section 5 would better fit in another "ablation study section" explaining each of the settings more thoroughly. I would also encourage the authors to explain possible reasons behind why the proposed method is able to outperform other baselines. For instance, `the impact of dropout', it is an interesting and useful study however digging further into reasons why it is performing better than without dropout would lead to more insights than empirically stating the results. Even for “Effect of the Number of Sampled Tokens”, I had to go through it multiple times to understand it is an ablation study for ‘m’ parameter. (d) In the introduction, it is not clearly explained what is meant by “multiple token embeddings.”

2) The notations in the methods section require careful attention. For example, in section 3, the authors introduce p_i as a linear combination of m embedding vectors weighted by p’_i, and they mention the dimension of E as e X m. It seems that the authors propose to randomly sample m embedding vectors for soft prompt initialization. However, in the subsequent step, they reparameterize E as follows: E = Efreeze + ∆E, with ∆Einit = 0e×n. The variable 'n' is not defined or mentioned anywhere prior to this point. Furthermore, the subsequent step, 'p_i = (Efreeze + ∆Ei)p′_i,' appears to contradict the dimensions mentioned earlier, as E was defined as having a dimension of e X m.

3) The authors deliberately choose not to include IPT and ATTEMPT in their comparative analysis, explaining that these methods depend on pretrained source prompts. This is at odds with their primary goal, which is to enhance soft prompt tuning without the need for pre-trained soft prompts. However, the rationale behind excluding these methods from the comparison is not entirely transparent. To bolster the validity of their approach, incorporating a comparison with these baseline methods would lead to more insights on the utility of this method. I would like to point out that utilizing the pretrained soft-prompts has a similar analogy as utilizing the pretrained weights for finetuning on a downstream task. I do not see a limitation in directly utilizing the pretrained soft prompts.

4) While the method is straightforward to implement, the authors have not sufficiently justified its apparent stability compared to other baseline methods for soft prompt tuning. For example, it remains unclear why the linear combination of 'm' randomly sampled token embeddings yields enhanced performance over the baseline. What specific aspects of this initialization contribute to superior results? The authors should provide additional insights to clarify this matter. A more comprehensive analysis is also needed to address the issue of inference without 'dropout' as mentioned earlier.

**Questions:**

1) I am not convinced with the result that, Superpos PT without dropout outperforms full finetuning by such a large margin of 16.4% on CB task on T5 small, same goes with T5 base on COPA task. It is not convincing that PEFT method is able to beat full finetuning by such a large margin. As claimed by the authors even LoRA and adapters that perform better than soft prompt based methods, occasionally beat full finetuning by very small margin. Could you please shed some light on it? Also, are the hyperparameters used to tune full model parameters consistent with the original paper.

2) The authors mention this in the introduction “Soft prompt is renowned for its exceptional parameter efficiency.” However it also mentions “finetuning soft prompts is optimization-intensive, particularly with limited data and smaller model sizes in T5 family between 50 to 300 million parameters (Lester et al., 2021);” As far as I understand the model weights are frozen. Are they contradictory statements made independently or are the authors mentioning that the convergence rate is slow for soft prompt based methods?

---

> ### Author Response · Authors · 2023-11-23
> **Response to the Reviewer gZe6**
>
> Dear Reviewer,
>
> Thank you for your valuable feedback. We have carefully considered your comments and made the following revisions to our manuscript. To prevent redundancy in the revised manuscript regarding additional experiments and analyses, please refer to 'Response to Reviewer 4x9F' for details on the changes in the following areas:
> 1. Expansion of the Evaluations
> 2. Stability Analysis of the Results
> 3. Enhanced Baseline Comparisons
> 4. Additional Experiments with T5-Base
> 5. Visual Analysis of Task Similarity
> 6. Superposition vs. Softmax Analysis
> 7. GPT-3.5-Turbo Inclusion.
>
> Regarding specific concerns:
>
> **Enhanced Comparative Analysis:** Our revised manuscript now includes a more comprehensive analysis, demonstrating the stability and performance of our method compared to baseline methods. We refer you to the 'Response to Reviewer 4x9F' on “1. Expansion of the Evaluations” and “Stability Analysis of the Results”.
>
> **IPT Exclusion Rationale:** We acknowledge the importance of comparing our method with IPT. However, IPT uses BART as its backbone, in contrast to our T5-based approach. A direct comparison would necessitate training an autoencoder on T5 prompts over numerous source tasks, a condition we couldn't replicate due to limited access to an adequate number of source prompts. We have emphasized this methodological limitation in our manuscript.
>
> **Question on Performance Margin:** Regarding the significant performance margin in smaller datasets like CB and COPA, we attribute this to the reduced risk of overfitting in PEFT methods compared to full finetuning. Full finetuning on such small datasets often leads to overfitting, while PEFT methods, by modifying a limited subset of parameters, mitigate this risk.
>
> **Terminology and Notation Clarifications:** We thank the reviewer for the feedback. The notation inconsistencies in Section 3 are solved and clarifying statements are provided (page 3).
>
> Question on **Parameter Efficiency and Optimization Intensity:** The statement about parameter efficiency refers to the effectiveness of soft prompts in adapting large models like T5 with fewer parameters. However, optimizing these prompts, especially in limited data scenarios, requires more training steps or epochs, thus resulting in a slower convergence rate, a trade-off that soft prompt tuning is dealing with.
>
> **Separate Ablation Study Section for Number of Sampled Tokens**: We have reorganized Section 5 and moved ‘Effect of the Number of Sampled Tokens’ to 'ablation study' in section 4.
> We appreciate your suggestion regarding the structure of the Introduction and Related Works sections. However, we believe that the inclusion of related work in the introduction is largely a matter of preference. Given this, we prefer to maintain the original structure, as it effectively presents the context and relevance of our work within the field.
>
> We hope that the discussed revisions effectively address your concerns. We are grateful for your guidance and remain open to any further suggestions.

---

### Official Review · Reviewer_4x9F · 2023-11-01

**Soundness:** 2 fair
**Presentation:** 3 good
**Contribution:** 2 fair
**Rating:** 5
**Confidence:** 4

**Summary:**

This paper focuses on soft prompt tuning of pretrained language model and proposes an approach called SuperPos-Prompt, which tunes the delta value of word embedding parameter as well as the combination weight of words for the soft prompt tokens. The authors have compared the proposed SuperPos-Prompt and existing prompt tuning methods, including Intrinsic Prompt Tuning, Residual Prompt Tuning and ATTEMPT, both theoretically and experimentally. The experimental results using the pretrained T5 model on GLUE and SuperGLUE benchmarks show the advantage of SuperPos-Prompt over existing methods. Meanwhile, as an additional contribution, the authors claim that the dropout should be omitting from the frozen network.

**Strengths:**

1. The proposed method is relatively easy to implement, compared with previous methods requiring heavily designed soft prompt initialization.
2. The selected setting of benchmark and pretrained model (T5) is representative.
3. Analysis of key components, including the impact of dropout, the results on different PLMs and effect of the number of sampled tokens are conducted.

**Weaknesses:**

1. The paper lacks sufficient analysis of SuperPos-Prompt learned soft prompts on what semantic information the prompt tokens indicate and how it contributes the results, at least some case study is needed.
2. To be honest, the novelty is fair and incremental. In my opinion, the paper is more like an experimental report over existing methods. If more analysis on the learned soft prompts from semantic perspective is proposed, I think it will be much better.

**Questions:**

Have you considered more recent LLMs and includes more baselines like few-shot results on ChatGPT/GPT4 for reference?

---

> ### Author Response · Authors · 2023-11-23
> **Response to the Reviewer 4x9F**
>
> Dear Reviewer,
>
> Thank you for your insightful feedback, which has significantly enhanced the quality of our article. In response to your comments, we conducted additional experiments and analyses. The changes fall into the following categories:
>
> **Expansion of the Evaluations:** We have extended our evaluation from seven NLP tasks to thirteen tasks, providing a broader analysis of different methods. The updated results are provided in Table 1.
>
> **Stability Analysis of the Results:** To address the reviewer’s concern regarding the stability of the results compared to other approaches, analogous to what is done in KoLA (Jifan Yu et al., 2023), we standardized the task scores and measured the mean and standard deviation across all tasks. We have added a new section on stability analysis (Section 5. Results - Stability Analysis), which includes a discussion on 'standardized overall scoring', along with a corresponding table (Table 2). In this analysis, we show that SuperPosPrompt achieved the highest overall score in comparison with state-of-the-art prompt tuning approaches while having the lowest standard deviation. SuperPosPrompt performs competitively compared to full fine-tuning in both overall score and variance.
>
> **Enhanced Baseline Comparisons:** In addressing the raised comment, we have included the ATTEMPT baselines as well. As we showed in updated Table 1, our method has superior performance on most tasks that ATTEMPT were tested on.
>
> **Additional Experiments with T5-Base:** We conducted additional experiments using the T5-Base checkpoint, allowing for direct comparisons with ATTEMPT and Residual Prompt Tuning.
>
> **Visual Analysis of Task Similarity:** Regarding the analysis of SuperPos-Prompt from a semantic perspective, we agree that this is an important aspect. In the current revision, we've included a preliminary visual analysis for task similarities. However, a more comprehensive semantic dissection requires sophisticated methodologies which we aim to explore in our future work.  In the revised paper, we introduced a visual analysis of task similarities using SuperPos-Prompt weights in Figure 2.d, providing a clearer understanding of how taks prompts are related to each other.
>
> **Superposition vs. Softmax Analysis:** In the appendix, we present a comparative study showing the superiority of superposition over softmax (Table A3).
>
> **Analysis of GPT-3.5:** In response to the comment about including OpenAI LLMs, we have added an analysis of GPT-3.5's one-shot performance in the appendix. GPT-3.5 achieved an average score of 67.1, while our approach achieved an average score of 81.2 (Table A3).
>
> We have conducted further professional proofreading to enhance the paper's readability. We hope these revisions, along with the substantial additional experiments and analyses, adequately address your concerns. We appreciate your guidance in strengthening our article.

---

### Meta-Review · Area_Chair_SRJg · 2023-12-02

**Metareview:**

The paper introduces "SuperPos-Prompt," an approach for soft prompt tuning in pre-trained language models. This method performs better than existing prompt tuning techniques when tested on GLUE and SuperGLUE benchmarks using the pre-trained T5 model.

Several reviewers have suggested that the paper could improve by providing more extensive comparisons with similar methodologies. Consequently, the results of ATTEMPT have been added to the revised version of the paper. Additionally, a more in-depth ablation study is recommended to better understand the factors contributing to SuperPos-Prompt's superior performance. The main distinction between ATTEMPT and SuperPos-Prompt lies in their approaches to initializing embeddings for the weighted sum. Exploring different initialization methods for SuperPos-Prompt could be insightful, helping to ascertain whether its enhanced performance is due to superior embedding initialization or other factors.

**Justification For Why Not Higher Score:**

The recommendation for rejection appears to be based on a combination of factors, despite the paper's introduction of an innovative approach in soft prompt tuning with "SuperPos-Prompt" and its demonstrated performance improvements:

1. Insufficient Comparative Analysis: A primary concern is the lack of extensive comparisons. While the revised version of the paper included results from ATTEMPT, the reviewers suggest that broader experiments with more pre-trained models.

2. Need for More In-Depth Ablation Study: The reviewers highlighted the need for a more detailed ablation study to dissect the factors contributing to the method's effectiveness.

3. Question of Methodological Novelty: There is a concern regarding the novelty of the approach. The proposed method is almost the same as ATTEMPT, simply with different embedding initialization.

4. Presentation: Although not explicitly stated in the meta-review, a reviewer's comments suggested issues with the presentation.

**Justification For Why Not Lower Score:**

N/A

---

### Decision · Program_Chairs · 2024-01-16

Reject